# Prussian Blue Nanoparticle Supported MoS_2_ Nanocomposites as a Peroxidase-Like Nanozyme for Colorimetric Sensing of Dopamine

**DOI:** 10.3390/bios12050260

**Published:** 2022-04-20

**Authors:** Zhiqiang Zhu, Lingbo Gong, Xiangyang Miao, Chaoyang Chen, Shao Su

**Affiliations:** 1Suzhou Chien-Shiung Institute of Technology, 1 Jianxiong Road, Suzhou 215411, China; zhuzq@csit.edu.cn (Z.Z.); miaoxy@csit.edu.cn (X.M.); chency@csit.edu.cn (C.C.); 2State Key Laboratory of Organic Electronics and Information Displays & Jiangsu Key Laboratory for Biosensors, Institute of Advanced Materials (IAM), Nanjing University of Posts and Telecommunications, 9 Wenyuan Road, Nanjing 210023, China; gonglingboo@163.com

**Keywords:** colorimetric sensor, dopamine, Prussian blue nanoparticles, molybdenum disulfide

## Abstract

An abnormal level of dopamine (DA) is usually related to neurological disorders, including Parkinson’s disease. Herein, cubic-shaped, Prussian blue nanoparticle-supported MoS_2_ nanocomposites (MoS_2_-CPBNPs) were prepared as peroxidase-like nanozymes for the label-free, colorimetric detection of DA. As expected, the as-prepared MoS_2_-CPBNPs nanozymes have outstanding peroxidase-like mimicking activity, which can catalyze 3,3′,5,5′-Tetramethylbenzidine (TMB) to generate blue, oxidized TMB in the presence of hydrogen peroxide (H_2_O_2_). DA can inhibit the oxidation of TMB, which causes blue solutions to fade and become colorless. According to this phenomenon, the developed colorimetric sensor can qualitatively and quantitatively analyze DA ranging from 0 to 300 μM with a detection limit of 0.09 μM. In addition, the high recovery and low relative standard deviation for practical DA determination suggested that this colorimetric sensor has potential for application in biological biosensing and diagnostic fields.

## 1. Introduction

When the intrinsic enzyme mimicking the activity of ferromagnetic nanoparticles was first discovered by Yan and co-workers [1], nanomaterials with enzyme-like characteristics became known as “nanozymes”. Nanozymes are emerging artificial enzymes that exhibit outstanding catalytic activity, high chemical and thermal stability, easy preparation and long-term storage [2,3]. On the basis of their advantages, nanozymes have been considered favorable alternatives to natural enzymes for sensors, disease diagnosis, imaging and therapy [4,5]. Among all applications, the development of colorimetric sensors is a particularly promising application of nanozymes due to its simplicity and the fact that it does not require equipment assistance. For example, Wei and co-workers developed a heteroatom-doped, graphene-based, colorimetric sensing platform for pesticide detection [6]. Utilizing the changes in the catalytic activity of graphene, the designed colorimetric sensing arrays could detect five pesticides with high performance. Though great advances in nanozymes have been achieved, it is still necessary to develop highly efficient nanozymes for high-performance chemical and biological molecule detection.

Nowadays, many nanomaterials have been known to exhibit enzyme-like activity, including noble metal nanoparticles [7], metal oxides [8], carbon nanotubes [9], graphene and derivatives [10], metal–organic framework [11,12,13], etc. Molybdenum disulfide (MoS_2_) is a graphene-like, two-dimensional layered nanomaterial that has been receiving increasing interest because of its unique structure and properties [14,15]. It has been reported that MoS_2_ nanosheets demonstrate peroxidase-like activity, which can be used to construct colorimetric sensors for chemical and biological molecule detection [16,17,18,19]. An efficient way to increase this enzyme-like activity is to form MoS_2_-based nanocomposites by hybridizing MoS_2_ nanosheets with functional groups. For example, Nirala et al. synthesized a nanocomposite composed of MoS_2_ nanoribbons and gold nanoparticles, which exhibited fascinating peroxidase-like activity. The wide pH and temperature tolerance led to the nanocomposites having excellent analytical performance for cholesterol detection [20]. Inspired by the above exciting findings, synthesis of MoS_2_-based nanocomposites is an efficient method of developing nanozymes with high enzyme-mimicking activity.

Herein, cubic-shaped, Prussian blue nanoparticles (CPBNPs) with high enzyme-mimicking activity were introduced during the construction of MoS_2_-based nanozymes [21,22]. Taking dopamine (DA) as an analytical model, a colorimetric sensor was designed based on CPBNPs-decorated MoS_2_ nanocomposites (MoS_2_-CPBNPs). It is well-known that DA is a neurotransmitter that plays a significant role in the function of metabolism, the cardiovascular system and the central nervous system. An abnormal DA level is often related to drug addiction and neurological disorders such as Parkinson’s disease [23,24]. As expected, the synthesized MoS_2_-based nanozymes exhibited excellent peroxidase-like properties and showed stronger affinity to 3,3′,5,5′-Tetramethylbenzidine (TMB) compared with natural horseradish peroxidase (HRP), generating a blue solution in the presence of hydrogen peroxide (H_2_O_2_). This is because MoS_2_-based nanozymes catalyze H_2_O_2_ to generate hydroxyl radical (·OH), which is often reacted with TMB to generate oxidized TMB with a blue color [25,26]. With the addition of DA, the solution color gradually faded, and the adsorption intensity decreased because DA consumed the hydroxyl radical. On the basis of this concept, the developed colorimetric sensor could determine DA in buffers and in real samples with satisfactory results (Figure 1). All data suggested that the as-prepared MoS_2_-CPBNPs nanozymes possess great potential for application in the construction of colorimetric sensors for chemical and biological molecule detection.

## 2. Materials and Methods

### 2.1. Chemicals and Reagents 

Molybdenum (IV) sulfide (MoS_2_, <2 mm, 99%), n-butyllithium, 3,3′,5,5′-Tetramethylbenzidine (TMB), dopamine (DA), glutathione (GSH) and uric acid (UA) were purchased from Sigma-Aldrich Ltd. Poly(N-vinyl-2-pyrrolidone) (PVP, MW = 58000), potassium chloride (KCl), potassium ferricyanide (K_3_[Fe(CN)_6_], ≥99.5%), ferric chloride (FeCl_3_), sodium acetate (NaAc), glacial acetic acid (HAc), 30% hydrogen peroxide (H_2_O_2_), ascorbic acid (AA), glucose (Glc), tyrosine (Tyr), serine (Ser), histidine (His), tryptophan (Trp), cysteine (Cys) and phenylalanine (Phe) were purchased from Sinopharm Chemical Reagent Co., Ltd. Other chemicals were of analytical grade without further purification. All aqueous solutions were prepared with twice-deionized water.

### 2.2. Apparatus

A microwave system (CEM Explorer) was employed to synthesize MoS_2_-based nanozymes. A Shimadzu UV-3600 spectrophotometer was used for measuring the absorption spectrum. Transmission electron microscope (TEM) images were taken on a Hitachi H-7500 electron microscope at an accelerating voltage of 120 kV.

### 2.3. Preparation of MoS_2_-CPBNPs Nanozymes

According to the published works [17,27], MoS_2_ nanosheets were prepared via classical lithium intercalation method. MoS_2_ nanosheets were centrifuged and purified to synthesize MoS_2_-CPBNPs nanozymes. First, 1000 μL PVP, 0.05 mM K_3_[Fe(CN)_6_] and 0.1 M KCl solution were added into 0.025 mg mL^−1^ MoS_2_ solution. Then, the solution pH was adjusted to 1.5 with hydrochloric acid. After that, 0.05 mM FeCl_3_ was added to the above homogeneously-mixed solution and stirred for 10 min. Finally, the homogeneous solution was reacted at 60 °C for 30 min with microwave-assisted method. The purified product was diluted in 1 mL ultrapure water and stored in a 4 °C refrigerator after purification.

### 2.4. Peroxidase-Like Activity of MoS_2_-CPBNPs Nanozymes

The enzyme-mimicking activity of MoS_2_-based nanozymes was evaluated by TMB + H_2_O_2_ reaction strategy. The effects of reaction time (1~15 min), pH value (3~8) and reaction temperatures (4~50 °C) on the catalytic activity were evaluated by monitoring the intensity of absorption peak at 652 nm. The steady-state kinetic parameters of MoS_2_-based nanozymes were studied by tuning the concentrations of H_2_O_2_ and TMB substrates.

## 3. Results and Discussion

### 3.1. Characterization of MoS_2_-CPBNPs Nanozymes

A transmission electron microscope (TEM) was utilized to prove the successful synthesis of MoS_2_-CPBNPs nanozymes. As shown in Figure 1A, the MoS_2_ nanosheet exhibited a typical layered nanostructure with some wrinkles. After decoration, uniform, cubic-shaped, Prussian blue nanoparticles with a diameter of about 105 nm were heavily supported on the surface of MoS_2_ nanosheets, forming MoS_2_-CPBNPs nanocomposites (Figure 1B). The solution color also proved the synthesis of MoS_2_-CPBNPs nanocomposites. As recorded in the inset photos, the solution color turned from brown-yellow to dark cyan after the decoration of CPBNPs, suggesting that MoS_2_-CPBNPs nanocomposites were successfully prepared.

After synthesis, the peroxidase-like catalytic behavior of MoS_2_-CPBNPs nanozymes was investigated by using TMB + H_2_O_2_ reaction strategy. As shown in Figure 1C, solutions containing TMB or MoS_2_-CPBNPs + TMB appeared colorless, indicating that no catalytic reaction occurred. Correspondingly, no obvious adsorption peaks were observed at 652 nm. Compared with TMB + H_2_O_2_ solutions, the addition of MoS_2_-CPBNPs nanozymes caused obvious catalytic behavior, accompanied by bluer solutions and higher adsorption peaks. It should be noted that MoS_2_ nanosheets and CPBNPs showed good peroxidase-like activity towards the oxidation of TMB in the presence of H_2_O_2_ (Figure 1D). Interestingly, the synergistic effect led to the developed nanozymes exhibiting better catalytic activity than single MoS_2_ nanosheets and CPBNPs nanozymes, respectively, resulting in higher adsorption peak intensity and deeper solution color. All experimental results indicate that MoS_2_-CPBNPs nanozymes have the potential to construct colorimetric sensors due to their excellent peroxidase-like properties.

### 3.2. Optimization of Experimental Condition

To obtain the best catalytic behavior, the effects of experimental conditions, such as reaction time, pH value and reaction temperature, on catalytic activity were studied. Figure 2A shows the relationship between the reaction time and the adsorption peak intensity of oxidized TMB (oxTMB). It can be seen that the peak intensity increased when the increasing reaction time ranged from 3 to 12 min and almost reached a plateau when the reaction time was 12 min. The solution color also showed no obvious changes when the reaction time ranged from 12 to 15 min. The inset photo of the solution color changes also supported this conclusion. Therefore, 12 min was chosen as the optimal reaction time. Similarly, the pH value and reaction temperature were also optimized. According to the experimental results recorded in Figure 2B and Figure 2C, the optimal pH value and reaction temperature were selected as 4 and 25 °C, respectively.

### 3.3. Kinetic Investigation of MoS_2_-CPBNPs Nanozymes

Steady-state kinetics were investigated to better understand the peroxidase-like activity of the as-prepared MoS_2_-CPBNPs nanozymes by tuning the concentration of TMB and H_2_O_2_. As recorded in Figure 3A,B, the catalytic responses of MoS_2_-CPBNPs nanozymes in different concentrations of TMB and H_2_O_2_ fit the typical Michaelis–Menten model, respectively. Meanwhile, Lineweaver–Burk curves (Figure 3C,D) were obtained by keeping one substrate concentration unchanged while varying the other. According to the experimental results, the Michaelis–Menten constant (K_m_) and the maximum reaction velocity (V_max_) of MoS_2_-CPBNPs nanozymes were evaluated by fitting the Lineweaver–Burk equation in double reciprocal plots, which are listed in Table 1. The K_m_ of MoS_2_-CPBNPs nanozymes for TMB and H_2_O_2_ were estimated as 0.22 and 3.17 mM, respectively, which are smaller than those of the HRP enzyme (H_2_O_2_: 3.7 mM and TMB: 0.43 mM) [1]. These exciting results suggested that MoS_2_-CPBNPs nanozymes have a much stronger affinity to TMB and H_2_O_2_ than HRP, further proving the excellent peroxidase-like activity of MoS_2_-CPBNPs nanozymes.

### 3.4. Colorimetric Detection of Dopamine Based on MoS_2_-CPBNPs Nanozymes

On the basis of excellent peroxidase-like activity, a colorimetric sensor based on MoS_2_-CPBNPs nanozymes was developed for DA detection. As shown in Figure 4A, the solution color faded after the addition of DA. Correspondingly, the adsorption peak intensity of oxTMB at 652 nm decreased, suggesting that the catalytic oxidation of TMB was inhibited by DA. The reason for this was ascribed to the fact that the generated hydroxyl-free radical was quenched by DA [28]. According to this phenomenon, the analytical performance of the as-developed MoS_2_-based colorimetric sensor was studied. Obviously, the absorption peak intensity decreased, and the solution color changed from blue to colorless, with the increasing DA concentration ranging from 0 to 300 μM (Figure 4B). A linear range was obtained between peak intensity and 1–100 μM DA, with a low detection limit of 0.09 μM (Figure 4C).

The performance of the MoS_2_-based colorimetric sensor was comparable to or better than previously reported works [29], which are recorded in Table 2. The excellent peroxidase-like activity of the MoS_2_-CPBNPs nanozyme gave this colorimetric sensor a wide linear range and a low detection limit for the detection of DA, indicating that the as-prepared MoS_2_-CPtNPs nanozyme was an ideal nanomaterial to construct colorimetric sensors for target molecule detection.

### 3.5. The Selectivity of the Colorimetric Sensor

To evaluate the selectivity of this sensor for DA detection, several interfering compounds, including uric acid (UA), ascorbic acid (AA), tyrosine (Tyr), serine (Ser), histidine (His), tryptophan (Trp), phenylalanine (Phe) and glucose (Glc), were selected. After adjusting the concentration of interfering compounds to be 10 times the DA concentration, the analytical performance of the developed colorimetric sensor for DA and interfering compounds’ detection was listed in Figure 5A. The peak intensity change ΔA = A_blank_ − A (A_blank_ and A represent the absorbance intensity of TMB + H_2_O_2_ reaction strategy in the absence and presence of DA or interfering compounds, respectively) suggested that other compounds have almost no significant interfering effect on the detection of DA (Figure 5B). Experimental results revealed that this colorimetric sensor has excellent selectivity for DA detection.

### 3.6. Determination of Dopamine in Real Samples

The practical detection ability of this colorimetric sensor was tested in a diluted serum by using the standard addition method. Accepted relative standard deviation (RSD, <5%) and recovery (98–102%) were obtained for different concentrations of DA detection, which are listed in Table 3. These results suggested that the prepared MoS_2_-based colorimetric sensor could be applied to dopamine determination in real samples.

## 4. Conclusions

In summary, MoS_2_-CPBNPs nanozymes with outstanding peroxidase-like activity were developed to construct a colorimetric sensor for DA detection. The lower K_m_ value revealed that the designed nanozymes have a stronger affinity to TMB and H_2_O_2_ than natural HRP enzyme, which implies that the developed colorimetric sensor displays high performance in regard to DA detection. The colorimetric sensor’s excellent anti-interference ability also shows promising potential for DA determination in practical samples. To the best of our knowledge, this is the first instance in which MoS_2_-CPBNPs nanozymes were used as an excellent alternative artificial enzyme to construct a colorimetric sensor, which may pave the way for the future construction of sensors for target molecule detection.

## Data Availability

Not applicable.

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
