# Peer review of "Prussian Blue Nanoparticle Supported MoS2 Nanocomposites as a Peroxidase-Like Nanozyme for Colorimetric Sensing of Dopamine"

_biosensors, 2022, doi:10.3390/bios12050260_

Round 1

Reviewer 1 Report

The authors reported the detection of dopamine using cubic-shaped Prussian Blue/MoS2 nanocomposites as a Peroxidase-like nanozyme for colorimetric detection of Dopamine. The submission requires revision before further consideration addressing the following points:-

  1. The title should be revised to be clear and precise. Redundant words such as ‘Cubic-shaped’, ‘Nanocomposites’  should be removed.
  2. The role of each component in MoS2-CPBNPs i.e., MoS2 and CPBNPs should be investigated separately.
  3. Table 2 should be expanded. There are several nanomaterials that were reported as peroxidase-like nanozyme.
  4. References are missed.
  5. It is unclear why the activity decreased with the increase in the temperature, Figure 2C.
  6. The introduction should be improved. A general overview of different materials should be updated, including these References; Microchemical Journal 2021, 163, 105873; Analytical and Bioanalytical Chemistry 2021, 413, 4407–4416; Chemical Engineering Journal 2022, 433, 133597
  7. The language should be revised and typos should be corrected.

Minors

  1. Remove column 2 in Table 2 since all analyte is the same i.e. dopamine.

Reviewer 2 Report

This article describes the preparation of cubic-shaped Prussian blue nanoparticles-supported MoS2 nanocomposites (MoS2-CPBNPs) for label-free colorimetric detection of dopamine (DA). However, I believe that revision is necessary to maintain a high standard of publication for the Journal.

  1. What unique properties do cubic-shaped Prussian blue nanoparticles (CPBNPs) possess that aid in the formation of stable nanozymes? How does this nanozyme react with DA and how is it capable of providing specificity? The authors should conduct the selectivity test and document their findings.
  2. As is well known, 3,3',5,5'-Tetramethylbenzidine (TMB) and horseradish peroxidase (HRP) combine to form a blue solution when hydrogen peroxide (H2O2) is added. How do CPBNPs nanozymes and DA reactions generate H2O2? Authors should thoroughly explain their work with the aid of schematics.
  3. Why did the authors not use gold nanoparticles rather than MoS2? AuNPs may be more effective in colorimetric detection of DA. The authors should conduct similar experiments to compare sensing performance.
  4. Why was the pH of MoS2 adjusted to 1.5 using hydrochloric acid?
  5. In Fig. 1a and 1b, authors should also demonstrate the elemental composition of the synthesised solution using TEM-EDX.
  6. In this study, dopamine was detected using a colorimetric method. Additionally, authors should consult other optical methods for DA detection and compare their performance, such as “Development of Dopamine Sensor using Silver Nanoparticles and PEG- Functionalized Tapered Optical Fiber Structure, IEEE Transactions on Biomedical Engineering, Vol. 67, Issue 6, 1542 - 1547, 2019”
  7. There are numerous effective two-dimensional materials that are widely used in the development of biosensors. The authors should explain why they chose to include molybdenum disulfide (MoS2) in their proposed work. 
  8. In Fig. 3, the authors should include an error bar and validate the experimental results by conducting at least three sets of experiments.
  9. Authors must exercise extreme caution when conducting reusability, reproducibility, pH testing, and selectivity experiments.
  10. Additionally, authors should include a table comparing the proposed sensor's performance to that of comparable existing sensors in order to demonstrate the novel nature of the proposed work.

Round 2

Reviewer 1 Report

The authors addressed most of the comments and the revised version can be accepted.

Reviewer 2 Report

Satisfactory Revision.